# An Effective Sensor Deployment Scheme that Ensures Multilevel Coverage of Wireless Sensor Networks with Uncertain Properties

**DOI:** 10.3390/s20071831

**Published:** 2020-03-25

**Authors:** Yu-Ning Chen, Wu-Hsiung Lin, Chiuyuan Chen

**Affiliations:** Department of Applied Mathematics, National Chiao Tung University, Hsinchu 300, Taiwan; abc89051@gmail.com (Y.-N.C.); wuhsiunglin@nctu.edu.tw (W.-H.L.)

**Keywords:** sensor deployment, sensor coverage problem, network planning, probabilistic sensing model, wireless sensor network

## Abstract

The coverage problem is a fundamental problem for almost all applications in wireless sensor networks (WSNs). Many applications even impose the requirement of multilevel (*k*) coverage of the region of interest (ROI). In this paper, we consider WSNs with uncertain properties. More precisely, we consider WSNs under the probabilistic sensing model, in which the detection probability of a sensor node decays as the distance between the target and the sensor node increases. The difficulty we encountered is that there is *no* unified definition of *k*-coverage under the probabilistic sensing model. We overcome this difficulty by proposing a “reasonable” definition of *k*-coverage under such a model. We propose a sensor deployment scheme that uses less number of deployed sensor nodes while ensuring good coverage qualities so that (i) the resultant WSN is connected and (ii) the detection probability satisfies a predefined threshold $p_{\mathrm{th}}$, where $0<p_{\mathrm{th}}<1$. Our scheme uses a novel “zone 1 and zone 1–2” strategy, where zone 1 and zone 2 are a sensor node’s sensing regions that have the highest and the second highest detection probability, respectively, and zone 1–2 is the union of zones 1 and 2. The experimental results demonstrate the effectiveness of our scheme.

## 1. Introduction

A wireless sensor network (WSN) has various applications in health-care, smart home, security, environmental exploration, and the military [1,2]. A sensor node (or simply node) is the basic component of a WSN. A WSN usually consists of numerous nodes deployed in a region of interest (ROI). Two nodes can communicate with each other if each is within the transmission range of the other, in which case we say that there is a link between them or that they are neighbors. Each node is able to collect data and process information and communicate with neighboring nodes.

Among various issues in WSNs, the coverage problem and the connectivity problem have been regarded as crucial foundations because many applications rely on them. Surveys for the coverage problem can be found in [3,4]. A good sensor deployment strategy should consider both coverage and connectivity. Sensor deployment not only determines the cost of constructing the network but also affects how well the given ROI will be monitored. This paper assumes that each node’s sensing region is of a disk shape (see Figure 1a) and all nodes have the same sensing range $r_{s}$ and communication range $r_{c}$.

Let *u* be a location in the ROI, $s_{i}$ be a node in the WSN, and $d(u,s_{i})$ be the Euclidean distance between *u* and $s_{i}$. Most of the past researches use the binary sensing model [5,6,7], where nodes are assumed to be accurate in detecting targets within their sensing ranges. More precisely, under the binary sensing model (see Figure 1b for an illustration), the detection probability $p(u,s_{i})$ of *u* by $s_{i}$ is defined as
$$p\left(u,s_{i}\right)=\left\{\begin{array}{cc} 1 & \mathrm{if}\;d\left(u,s_{i}\right)\leq r_{s},\\ 0 & \mathrm{otherwise}. \end{array}\right.$$

In physical scenarios, the distance decay effect cannot be ignored. That is, the detection probability of a node decays as the distance between the target and the node increases. Based on this, the coverage problem under a more realistic model, called the probabilistic sensing model, has been investigated; see [7,8,9,10,11]. A survey for the coverage problem with uncertain properties can be found in [12]. Do notice that unlike the binary sensing model, which has a unified definition of $p(u,s_{i})$, there is no unified definition of $p(u,s_{i})$ under the probabilistic sensing model; see Section 2 for details. We now give the definition of $p(u,s_{i})$ used in [7] and in this paper. Under the probabilistic sensing model (see Figure 1c for an illustration), the detection probability $p(u,s_{i})$ of *u* by $s_{i}$ is defined as
(1)$$p\left(u,s_{i}\right)=\left\{\begin{array}{cc} e^{-\lambda d(u,s_{i})} & \mathrm{if}\;d\left(u,s_{i}\right)\leq r_{s},\\ 0 & \mathrm{otherwise}, \end{array}\right.$$
where λ is a sensor-dependent parameter.

Many real-world applications impose the requirement of multilevel k>1 coverage. For example, k≥2 for military or surveillance applications, k≥3 for positioning protocols using triangulation [13], conducting data fusion [14], and minimizing the impact of sensor failure [15]. Sensor deployment with multilevel (*k*) coverage have been discussed in [7,15,16,17].

Under the binary sensing model, an ROI is said to be *k*-*covered* if every location in the ROI can be detected by at least *k* nodes (i.e., every location in the ROI is within *k* nodes’ sensing regions). Unfortunately, to the best of our knowledge, there is no unified definition of *k*-coverage under the probabilistic sensing model. We now are ready to elaborate this issue. Let $A=\{s_{1},s_{2},\ldots ,s_{n}\}$ be the set of sensor nodes deployed in the ROI. In [7], a location *u* in the ROI is considered as *k*-covered if the probability that there are at least *k* nodes that can detect *u* is not smaller than a predefined threshold $p_{\mathrm{th}}$, where $0<p_{\mathrm{th}}<1$. More precisely, in [7], a location *u* in the ROI is said to be *k-covered* if
(2)$$\prod_{s_{i}\in A^{\prime }}p\left(u,s_{i}\right)\geq p_{\mathrm{th}},\;\mathrm{for}\;\mathrm{some}\;A^{\prime }\subseteq A\;\mathrm{with}\;\left|A^{\prime }\right|=k.$$

For clarity, we call this definition of *k*-coverage the ***k*-threshold coverage**. In [18], a location *u* in the ROI is said to be *k-covered* if
(3)$$\sum_{s_{i}\in A}p\left(u,s_{i}\right)\geq k.$$

For clarity, we call this definition of *k*-coverage the ***k*-expectation coverage**. In both [7] and [18], the ROI is said to be *k-covered* if every location *u* inside the ROI is *k*-covered. The $p(u,s_{i})$ used in [18] is defined as (Equation 4), which is a generalization of (Equation 1) and is introduced in Section 2. By (Equation 3), the event of *u* detected by $s_{i}$ is independent of the event of *u* detected by $s_{j}$ for j≠i. Reference [18] proves that the ROI is *k*-covered by A if, among all locations inside the ROI, the minimum expected number of nodes is *k* (there is a typo and *k* should be *at leastk k*).

Three main considerations in WSNs’ coverage are: maximizing coverage quality, maximizing network lifetime, and minimizing the number of deployed nodes. The coverage quality and lifetime are two conflict factors with respect to energy consumption. Reference [12] points out that the *k*-coverage requirement makes the problem more sophisticated because more nodes are needed; deploying less nodes by decreasing the overlap region and prolonging the network lifetime become complicated. The *k*-expectation coverage [18] has the drawback that the user cannot specify his/her preference for the threshold $p_{\mathrm{th}}$, meaning that the user cannot specify their desired coverage quality. Moreover, it is possible that the entire ROI is regarded as *k*-covered but some locations inside it are detected by a lot of nodes each with a small detection probability. The *k*-threshold coverage [7] has the drawback that it tends to use too many sensor nodes; we give the calculation details in Section 2. In short, Wang and Tseng [7] first calculate $r_{\mathrm{th}}$ (notice that $r_{\mathrm{th}}$ is denoted as $r_{s}^{p}$ in [7]) and then replace the original $r_{s}$ with $r_{\mathrm{th}}$ in the deployment.

The objective of this paper is to propose a sensor deployment scheme to use less number of nodes while ensuring the following two important coverage qualities: (i) the resultant WSN is connected and (ii) the detection probability satisfies a predefined threshold $p_{\mathrm{th}}$, where $0<p_{\mathrm{th}}<1$. Although *k*-threshold coverage [7] achieves the same objective, we find that the nature of *k*-threshold coverage makes $r_{\mathrm{th}}$ tend to become *much smaller than* the original $r_{s}$. For example, suppose k=3 and m denotes meters. Then:If λ=0.05 and $r_{s}=30$ m, then $r_{\mathrm{th}}$ for $p_{\mathrm{th}}$ = 0.7, 0.8, 0.9 are 2.377 m, 1.487 m, and 0.702 m, respectively.If λ=0.08 and $r_{s}=30$ m, then $r_{\mathrm{th}}$ for $p_{\mathrm{th}}$ = 0.7, 0.8, 0.9 are 1.486 m, 0.929 m, and 0.439 m, respectively.

These very small $r_{\mathrm{th}}$’s cause a large number of nodes to be deployed. Since the number of nodes is very large, the overall detection probability might be underestimated.

In this paper, we try to propose a “reasonable” solution to the *k*-coverage problem under the probabilistic sensing model. The main contributions of this paper are as follows.
We propose a “reasonable” definition of *k*-coverage under the probabilistic sensing model, called the ***k*-layer coverage**, and propose a scheme to achieve *k*-layer coverage.We propose a novel “zone 1 and zone 1–2” strategy to fulfill *k*-layer coverage scheme and ensure good coverage quality. We propose an efficient algorithm to calculate the radius $r_{1}$ of zone 1, which takes at most 18 iterations when error tolerance $\epsilon =10^{-6}$.Experimental results shows that our *k*-layer coverage scheme indeed uses less sensor nodes, thereby demonstrating the effectiveness of our scheme.

Our *k*-layer coverage scheme partitions the nodes into *k* subsets, each forming one layer of coverage, and ensures that for every location *u* inside the ROI, the detection probability for *u* by nodes in “each layer” is not smaller than $p_{\mathrm{th}}$. In particular, our *k*-layer coverage scheme ensures that every location *u* inside the ROI is within zone 1 of at least *k* nodes and within zone 1–2 of at least another 2·k nodes (zone 1 and zone 1–2 are defined in Section 4). When coverage quality is considered, our *k*-layer coverage scheme provides a good solution. Different from the *k*-threshold coverage [7], which replaces the original $r_{s}$ with $r_{\mathrm{th}}$, our *k*-layer coverage replaces the original $r_{s}$ with $r_{1}$. We prove that as long as $\sqrt{3}r_{1}<r_{s}$, we have $r_{1}>k\cdot r_{\mathrm{th}}$, meaning that our *k*-layer coverage will use less nodes. When the number of nodes is considered, our *k*-layer coverage scheme also provides a good solution.

The rest of this paper is organized as follow. Section 2 introduces the related works. Section 3 gives preliminaries, assumptions, and objectives. Section 4 gives the basics of the *k*-layer coverage. Section 5 gives our *k*-layer coverage scheme. Section 6 illustrates experimental results. Concluding remarks are given in the final section.

## 2. Related Works

Recall that a good sensor deployment should consider both coverage and connectivity. Assuming that the ROI is a convex set, Zhang and Hou [19] investigate the relationship between coverage and connectivity and prove that $r_{c}\geq 2r_{s}$ is both necessary and sufficient to ensure that coverage implies connectivity. With such a proof, one can then focus only on the coverage problem. As long as $r_{c}\geq \sqrt{3}r_{s}$, nodes can be deployed according to the regular triangular lattice pattern (triangular pattern for short; see Figure 2) so that both coverage and connectivity can be ensured [7,20]. Using the triangular pattern, neighboring nodes will be regularly separated by a distance of $\sqrt{3}r_{s}$. A deployment using the triangular pattern is sometimes called an *optimal deployment* since it is asymptotically optimal in terms of the number of nodes needed to achieve full coverage of the ROI.

In some applications, $r_{c}\geq 2r_{s}$ or $r_{c}\geq \sqrt{3}r_{s}$ may not hold. Wang and Tseng [7] therefore consider an arbitrary relationship between $r_{s}$ and $r_{c}$, thus relaxing the limitations of existing results. In [7], for the the binary sensing model, two solutions to achieve *k*-coverage are proposed: the naive duplicate placement scheme (duplicate scheme for short) and the interpolating placement scheme (interpolating scheme for short). The idea of the duplicate scheme is to use a good sensor placement method to ensure 1-coverage and connectivity and then duplicate *k* nodes on each designated location. However, since the duplicate scheme may result in some regions in the ROI having a much higher coverage levels than *k*, the interpolation scheme is therefore being proposed to “reuse” these regions to generate a multilevel coverage. When k=1 or k=2 or (k≥3 and $r_{c}>\frac{2+\sqrt{3}}{3}r_{s}$), it is found that the interpolation scheme will not save nodes compared to the duplicate scheme, thereby adopting the duplicate scheme. For clarity, we summarize the schemes used in [7] in Table 1. Notice that the interpolation schemes used in the ($r_{c}\leq \frac{\sqrt{3}}{2}r_{s}$)-case and the ($\frac{\sqrt{3}}{2}r_{s}<r_{c}\leq \frac{2+\sqrt{3}}{3}r_{s}$)-case are different.

Wang and Tseng [7] adapt the schemes of the binary sensing model *to* the probabilistic sensing model. Set (★) = (k≥3 and $r_{c}\geq \sqrt{3}r_{s}$) for convenience. We now use the (★)-case as an illustration to show how [7] performs the adaption. According to Table 1 (shown in Section 2), under the binary sensing model, [7] will use the duplicate scheme and triangular pattern in the (★)-case. Under the probabilistic sensing model, in the (★)-case, [7] first calculates $r_{\mathrm{th}}$ (notice that $r_{\mathrm{th}}$ is denoted as $r_{s}^{p}$ in [7]). Then, [7] replaces the original $r_{s}$ with $r_{\mathrm{th}}$ in the deployment to ensure that every location inside the ROI is *k*-covered under the probabilistic sensing model. Since the duplicate scheme places *k* nodes on each designated location, [7] calculates $r_{\mathrm{th}}$ by
$$\prod_{s_{i}\in A^{\prime }}p\left(u,s_{i}\right)\approx e^{-k\lambda r_{\mathrm{th}}}$$
and
$$e^{-k\lambda r_{\mathrm{th}}}\geq p_{\mathrm{th}}\;\Rightarrow \;r_{\mathrm{th}}\leq \frac{\ln p_{\mathrm{th}}}{-k\lambda }$$
where *u* is a location in the ROI having the minimum *k*-covered probability and *u* is detected by a set $A^{\prime }$ of *k* nodes placed at a designated location with distance $r_{\mathrm{th}}$ to *u*.

Let *u*, $s_{i}$, $d(u,s_{i})$, and $p(u,s_{i})$ be defined as in Section 1. In [11,18,21,22], four parameters $\left(\lambda ,\beta ,r,r_{e}\right)$ are used to specify a probabilistic sensing model and $p(u,s_{i})$ is defined as
(4)$$p\left(u,s_{i}\right)=\left\{\begin{array}{cc} 1 & \mathrm{if}\;d\left(u,s_{i}\right)\leq r-r_{e},\\ e^{-\lambda {\left(d\left(u,s_{i}\right)-\left(r-r_{e}\right)\right)}^{\beta }} & \mathrm{if}\;r-r_{e}<d\left(u,s_{i}\right)\leq r+r_{e},\\ 0 & \mathrm{otherwise}, \end{array}\right.$$
where λ and β are sensor-dependent parameters (see Figure 3). That is, if $d\left(u,s_{i}\right)\leq r-r_{e}$, then a target at *u* will definitely be detected by $s_{i}$; if $d\left(u,s_{i}\right)>r+r_{e}$, then the detection probability will be too small and will be totally ignored. If $r-r_{e}<d\left(u,s_{i}\right)\leq r+r_{e}$, then the behavior of the detection probability obeys the function $e^{-\lambda {\left(d\left(u,s_{i}\right)-\left(r-r_{e}\right)\right)}^{\beta }}$. By taking β=1 and $r=r_{e}=\frac{r_{s}}{2}$, (Equation 4) coincides with (Equation 1).

Notice that besides the binary sensing model and the probabilistic sensing model, some researchers consider the evidence-based sensor coverage model and use the theory of belief functions to solve the coverage problem; see [23,24,25,26]. Notice that [23,24,25,26] consider 1-coverage and their ROI is assumed as a two- or three-dimensional grid of points. In this paper, we consider *k*-coverage and our ROI contains every location inside it. Some other references related to *k*-coverage can also be found in [27,28,29,30,31]. Before ending this section, for clarity, we summarize our most related works in Table 2.

## 3. Preliminaries, Assumptions, and Objectives

Typical types of coverage are target coverage, area coverage, and barrier coverage. The purpose of target coverage is to cover (monitor) a set of specific targets (or points), that of area coverage is to cover the entire ROI, and that of barrier coverage is to detect intruders who intend to cross a long belt region. In a WSN, nodes are deployed in ad-hoc or pre-planned manner. In pre-planned deployment, nodes are placed to designated locations in the ROI so that fewer nodes can provide satisfactory coverage with lower network maintenance and management cost. Area coverage is usually more difficult and can be solved by ad-hoc or pre-planned deployment, depending on the application scenario.

This paper discusses “area coverage” by “pre-planned deployment”; the following assumptions are made:The ROI is of a rectangular shape.Sensor nodes are homogeneous, i.e., with the same sensing range $r_{s}$ and communication range $r_{c}$.$r_{c}\geq \sqrt{3}r_{s}$.The application that we are interested in allows nodes to be deployed in pre-planned manner.The detection probability, $p(u,s_{i})$ of *u* by $s_{i}$, used in this paper is (Equation 1).

The objective of this paper is to propose a “reasonable” definition of *k*-coverage under the probabilistic sensing model and to develop a sensor deployment scheme that uses less number of nodes while ensuring the following two coverage qualities: (i) the resultant WSN is connected and (ii) the detection probability satisfies a predefined threshold $p_{\mathrm{th}}$, where $0<p_{\mathrm{th}}<1$.

## 4. Basics of the *k*-Layer Coverage

### 4.1. The Definition of *k*-Layer Coverage

**Definition** **1.**
*Given an ROI and a set A of a finite number of nodes deployed in the ROI, a location u in the ROI is k-layer covered (or k-covered for short) if A can be partitioned into k subsets $A_{1},A_{2},\ldots ,A_{k}$ such that*
(5)$$1-\prod_{s_{i,j}\in A_{i}}\left(1-p\left(u,s_{i,j}\right)\right)\geq p_{\mathrm{th}},\;\mathrm{for}\;\mathrm{each}\;i=1,2,\ldots ,k$$
*for a predefined threshold $p_{\mathrm{th}}$, where $0<p_{\mathrm{th}}<1$. The ROI is k-layer covered (or k-covered for short) if every location inside it is k-layer covered.*


For convenience, we call each $A_{i}$ one “layer” and call the above coverage the ***k*-layer coverage**. See Figure 4 for an illustration. The *k*-layer coverage ensures that the detection probability for *u* contributed by each layer (i.e., each $A_{i}$) is not smaller than $p_{\mathrm{th}}$. Thus, *u* is *k*-layer covered if and only if it is 1-layer covered by each $A_{i}$, i=1,2,…,k. Consequently, the ROI is *k*-layer covered if and only if it is 1-layer covered by each $A_{i}$, i=1,2,…,k.

### 4.2. “Zone 1 and Zone 1–2” Strategy

We regard each node’s sensing region as the composition of three concentric circular zones: zone 1, zone 2, and zone 3, as shown in Figure 5. We denote by $r_{1}$ the radius of zone 1 and $r_{2}$, the outer contour radius of zone 2, where $r_{1}<r_{2}\leq r_{s}$. We take $r_{2}=\sqrt{3}r_{1}$ so that the calculations can be greatly simplified. Zone 1 has a higher detection probability than zone 2 which further has a higher detection probability than zone 3. Zone 3 might be empty and our coverage scheme will not take it into account. For convenience, denote by zone 1–2 the union of zones 1 and 2.

The following theorem states the “zone 1 and zone 1–2” strategy used in our *k*-layer coverage scheme, whose proof is in Section 5.

**Theorem** **1.**
*Our k-layer coverage scheme ensures that for each $A_{i}$, every location inside the ROI is within zone 1 of one node in $A_{i}$ and within zones 1–2 of at least another two nodes in $A_{i}$.*


### 4.3. An Algorithm for Calculating the Radius $r_{1}$ of Zone 1

To use the “zone 1 and zone 1–2” strategy, we need to calculate $r_{1}$. Let *u* be an arbitrary location inside the ROI. By Theorem 1, there exist three distinct nodes $s_{a}$, $s_{b}$, and $s_{c}$ in $A_{i}$ such that *u* is within zone 1 of $s_{a}$ and zones 1–2 of $s_{b}$ and $s_{c}$. By (Equation 5), we have
$$\begin{array}{ccc} 1-\prod_{s_{i,j}\in A_{i}}\left(1-p\left(u,s_{i,j}\right)\right) & \geq & 1-\underset{\mathrm{node}\;s_{a}}{\underset{︸}{\left(1-p\left(u,s_{a}\right)\right)}}\underset{\mathrm{node}\;s_{b}}{\underset{︸}{\left(1-p\left(u,s_{b}\right)\right)}}\underset{\mathrm{node}\;s_{c}}{\underset{︸}{\left(1-p\left(u,s_{c}\right)\right)}}\\ & \geq & 1-\left(1-e^{-\lambda r_{1}}\right)\left(1-e^{-\lambda r_{2}}\right)\left(1-e^{-\lambda r_{2}}\right). \end{array}$$

On the one hand, to minimize the number of nodes, $r_{1}$ should be as large as possible. On the other hand, to ensure coverage quality, $r_{1}$ cannot be too large because the following inequality is needed:(6)$$1-\left(1-e^{-\lambda r_{1}}\right)\left(1-e^{-\lambda r_{2}}\right)\left(1-e^{-\lambda r_{2}}\right)\geq p_{\mathrm{th}}.$$

Therefore, we should find the largest possible $r_{1}$ that satisfies (Equation 6). The following algorithm finds such an $r_{1}$ and we now briefly describe the idea behind our algorithm.

Since the detection probability of any target outside $r_{s}$ is defined as zero, $r_{1}$ and $r_{2}$ must satisfy $r_{1}<r_{2}\leq r_{s}$. Since we take $r_{2}=\sqrt{3}r_{1}$, the largest possible $r_{1}$ is therefore $\frac{r_{s}}{\sqrt{3}}$. Substituting $r_{1}$ by $\frac{r_{s}}{\sqrt{3}}$ into (Equation 6), we have
$$1-\left(1-e^{-\lambda \frac{r_{s}}{\sqrt{3}}}\right)\left(1-e^{-\lambda r_{s}}\right)\left(1-e^{-\lambda r_{s}}\right)\geq p_{\mathrm{th}}.$$

Thus, our *k*-layer coverage has an unavoidable lower bound $p_{\mathrm{th}}^{\min}$ of $p_{\mathrm{th}}$, which is defined as
$$p_{\mathrm{th}}^{\min}=1-\left(1-e^{-\lambda \frac{r_{s}}{\sqrt{3}}}\right)\left(1-e^{-\lambda r_{s}}\right)\left(1-e^{-\lambda r_{s}}\right).$$

This unavoidable lower bound $p_{\mathrm{th}}^{\min}$ is due to the system assumption that the detection probability of any target outside $r_{s}$ is zero. If $p_{\mathrm{th}}\leq p_{\mathrm{th}}^{\min}$, then our algorithm *enlarges*
$p_{\mathrm{th}}$ to be $p_{\mathrm{th}}^{\min}$ and returns $\frac{r_{s}}{\sqrt{3}}$ for $r_{1}$. In the remaining part of this paragraph, we assume $p_{\mathrm{th}}>p_{\mathrm{th}}^{\min}$ and show how to calculate $r_{1}$ when $p_{\mathrm{th}}>p_{\mathrm{th}}^{\min}$. Since
$$1-{\left(1-e^{-\lambda r_{2}}\right)}^{3}\leq 1-\left(1-e^{-\lambda r_{1}}\right)\left(1-e^{-\lambda r_{2}}\right)\left(1-e^{-\lambda r_{2}}\right)\leq 1-{\left(1-e^{-\lambda r_{1}}\right)}^{3},$$
we have
(7)$$1-{\left(1-e^{-\lambda r_{2}}\right)}^{3}\leq p_{\mathrm{th}}\leq 1-{\left(1-e^{-\lambda r_{1}}\right)}^{3}.$$

For convenience, denote by $r_{1}^{*}$ the the largest possible $r_{1}$ that satisfies (Equation 6) under the assumption that $p_{\mathrm{th}}>p_{\mathrm{th}}^{\min}$. We say that a value is an *ϵ-approximation* of $r_{1}^{*}$ if it satisfies (Equation 6) and satisfies a given error tolerance ϵ. The following method finds either $r_{1}^{*}$ or an ϵ-approximation of $r_{1}^{*}$ by using (Equation 7). Substituting $\sqrt{3}r_{1}$ for $r_{2}$ in (Equation 7), we have
$$1-{\left(1-e^{-\lambda \sqrt{3}r_{1}}\right)}^{3}\leq p_{\mathrm{th}}\leq 1-{\left(1-e^{-\lambda r_{1}}\right)}^{3}.$$

Thus
$$1-e^{-\lambda r_{1}}\leq {\left(1-p_{\mathrm{th}}\right)}^{\frac{1}{3}}\leq 1-e^{-\lambda \sqrt{3}r_{1}}.$$

Hence
$${\left(e^{-\lambda r_{1}}\right)}^{\sqrt{3}}=e^{-\lambda \sqrt{3}r_{1}}\leq 1-{\left(1-p_{\mathrm{th}}\right)}^{\frac{1}{3}}\leq e^{-\lambda r_{1}}.$$

By using $1-{\left(1-p_{\mathrm{th}}\right)}^{\frac{1}{3}}\leq e^{-\lambda r_{1}}$, we can obtain an approximate $r_{1}$. Since $1-{\left(1-p_{\mathrm{th}}\right)}^{\frac{1}{3}}$ is a lower bound of $e^{-\lambda r_{1}}$, let
$$lower=1-{\left(1-p_{\mathrm{th}}\right)}^{\frac{1}{3}}.$$

By using $e^{-\lambda r_{1}}\leq {\left(1-{\left(1-p_{\mathrm{th}}\right)}^{\frac{1}{3}}\right)}^{\frac{1}{\sqrt{3}}}$, we can obtain another approximate $r_{1}$. Since ${\left(1-{\left(1-p_{\mathrm{th}}\right)}^{\frac{1}{3}}\right)}^{\frac{1}{\sqrt{3}}}$ is an upper bound of $e^{-\lambda r_{1}}$, let
$$upper={\left(1-{\left(1-p_{\mathrm{th}}\right)}^{\frac{1}{3}}\right)}^{\frac{1}{\sqrt{3}}}.$$

By keeping
$$lower\leq e^{-\lambda r_{1}}\leq upper,$$
when our algorithm stops, it finds either $r_{1}^{*}$ or an ϵ-approximation of $r_{1}^{*}$. Our full algorithm is now shown in Algorithm 1.
**Algorithm 1:****Input:**$p_{\mathrm{th}}$, $r_{s}$, and λ, where $p_{\mathrm{th}}$ is the threshold, $r_{s}$ is the sensing range, and  λ is the sensor-dependent parameter given in (Equation 1).
**Output:**
$\frac{r_{s}}{\sqrt{3}}$ if $p_{\mathrm{th}}\leq p_{\mathrm{th}}^{\min}$; $r_{1}^{*}$ or an ϵ-approximation of $r_{1}^{*}$ if $p_{\mathrm{th}}>p_{\mathrm{th}}^{\min}$.
1:**if**$p_{\mathrm{th}}\leq p_{\mathrm{th}}^{\min}$**then**$r_{1}\leftarrow \frac{r_{s}}{\sqrt{3}}$2:**else**3:    *found*← false;4:    $\epsilon \leftarrow 10^{-6}$;5:    *lower*
$\leftarrow 1-{\left(1-p_{\mathrm{th}}\right)}^{\frac{1}{3}}$;6:    upper
$\leftarrow {\left(1-{\left(1-p_{\mathrm{th}}\right)}^{\frac{1}{3}}\right)}^{\frac{1}{\sqrt{3}}}$;7:    **while** ((*found* = false) and (*upper* − *lower*) ≥ *ϵ*)) **do**8:        $mid\leftarrow \frac{lower+upper}{2}$;9:        $val\leftarrow 1-\left(1-mid\right)\left(1-mid^{\sqrt{3}}\right)\left(1-mid^{\sqrt{3}}\right)$;10:        **if**
$\left(val=p_{\mathrm{th}}\right)$
**then**
found← true;11:        **else if**
$\left(val<p_{\mathrm{th}}\right)$
**then**
lower←mid;12:        **else**
upper←mid;13:        **end if**14:    **end while**15:    **if** (found= true) **then**16:        $r_{1}\leftarrow \frac{\ln mid}{-\lambda }$;   //$r_{1}^{*}$17:    **else**18:        $r_{1}\leftarrow \frac{\ln upper}{-\lambda }$;   //an ϵ-approximation of $r_{1}^{*}$19:    **end if**20:**end if**21:return $r_{1}$;


**Theorem** **2.**
*If $p_{\mathrm{th}}\leq p_{\mathrm{th}}^{\min}$, then Algorithm 1 returns $r_{1}=\frac{r_{s}}{\sqrt{3}}$ in O(1) time. If $p_{\mathrm{th}}>p_{\mathrm{th}}^{\min}$, then Algorithm 1 returns either $r_{1}^{*}$ or an ϵ-approximation of $r_{1}^{*}$ for $\epsilon =10^{-6}$ after at most 18 iterations.*


**Proof.** The first statement is obvious. Assume that $p_{\mathrm{th}}>p_{\mathrm{th}}^{\min}$ and consider the second statement. Then, Algorithm 1 returns either $r_{1}=\frac{\ln\;mid}{-\lambda }$ or $r_{1}=\frac{\ln\;upper}{-\lambda }$. Consider the former case. In this case, $mid=e^{-\lambda r_{1}}$. By lines 9∼10, $1-\left(1-mid\right)\left(1-mid^{\sqrt{3}}\right)\left(1-mid^{\sqrt{3}}\right)=p_{\mathrm{th}}$. Thus $r_{1}$ is exactly $r_{1}^{*}$. Now, consider the latter case. In this case, $r_{1}<r_{1}^{*}$. Hence
$$1-\left(1-e^{-\lambda r_{1}}\right)\left(1-e^{-\lambda \sqrt{3}r_{1}}\right)\left(1-e^{-\lambda \sqrt{3}r_{1}}\right)>1-\left(1-e^{-\lambda r_{1}^{*}}\right)\left(1-e^{-\lambda \sqrt{3}r_{1}^{*}}\right)\left(1-e^{-\lambda \sqrt{3}r_{1}^{*}}\right)=p_{\mathrm{th}}.$$Thus $r_{1}$ satisfies (Equation 6). The execution of the while-loop ensures that $r_{1}$ satisfies the error tolerance ϵ. Therefore $r_{1}$ is an ϵ-approximation of $r_{1}^{*}$. We now prove that when $\epsilon =10^{-6}$, Algorithm 1 takes at most 18 iterations. Set $x=1-{\left(1-p_{\mathrm{th}}\right)}^{\frac{1}{3}}$ for easy writing. Since $0<p_{\mathrm{th}}<1$, we have 0<x<1. By line 5, initially lower=x. By line 6, initially $upper=x^{\frac{1}{\sqrt{3}}}$. Therefore, initially $upper-lower=x^{\frac{1}{\sqrt{3}}}-x$. Let $f\left(x\right)=x^{\frac{1}{\sqrt{3}}}-x$. The function f(x) achieves its maximum value when $f^{\prime }\left(x\right)=\frac{1}{\sqrt{3}}x^{\frac{\sqrt{3}-3}{3}}-1=0$, which occurs when
$$x={\sqrt{3}}^{\frac{3}{\sqrt{3}-3}}=3^{\frac{1}{2}\cdot \frac{3}{\sqrt{3}-3}\cdot \frac{\sqrt{3}-3}{\sqrt{3}+3}}=3^{\frac{3+\sqrt{3}}{-4}}.$$Note that $f(3^{\frac{3+\sqrt{3}}{-4}})≓0.199572<0.2$. Moreover, each iteration of the while-loop cuts the search space by half. Thus after the execution of *i*-th iteration of the while-loop,
$$\left(upper-lower\right)<f\left(3^{\frac{3+\sqrt{3}}{-4}}\right)\times 2^{-i}<0.2\times 2^{-i}.$$When $0.2\times 2^{-i}<\epsilon$ (i.e., when $i>\log_{2}\frac{0.2}{\epsilon }$), Algorithm 1 terminates its while-loop. Since $\epsilon =10^{-6}$, we have $\log_{2}\frac{0.2}{\epsilon }≓17.61<18$. Thus Algorithm 1 takes at most 18 iterations. □

The following corollary follows from the proof of the above theorem.

**Corollary** **1.**
*For an arbitrary error tolerance ϵ, Algorithm 1 takes at most $\left\lceil \log_{2}\frac{0.2}{\epsilon }\right\rceil$ iterations. In particular, when $\epsilon =10^{-x}$, where x is a positive integer, Algorithm 1 takes at most $\left\lceil \left(x-1\right)\log_{2}10+1\right\rceil$ iterations.*


Suppose m denotes meters. Algorithm 1 obtains:If λ=0.05, $r_{s}=30$ m, then $r_{1}$ for $p_{\mathrm{th}}=0.7,0.8,0.9$ are 15.685m, 12.391m, and 8.749m, respectively.If λ=0.08, $r_{s}=30$ m, then $r_{1}$ for $p_{\mathrm{th}}=0.7,0.8,0.9$ are 9.801m, 7.743m, and 5.468m, respectively.

To handle the probabilistic sensing model, Wang and Tseng [7] replace the original $r_{s}$ with $r_{\mathrm{th}}=\frac{\ln p_{\mathrm{th}}}{-k\lambda }$ in their deployment. To handle the probabilistic sensing model, we replace the original $r_{s}$ with $r_{1}$. The relationship between $r_{1}$ and $r_{\mathrm{th}}$ is given in the following lemma.

**Lemma** **1.**
*If $\sqrt{3}r_{1}<r_{s}$ and $1-\left(1-e^{-\lambda r_{1}}\right)\left(1-e^{-\lambda \sqrt{3}r_{1}}\right)\left(1-e^{-\lambda \sqrt{3}r_{1}}\right)=p_{\mathrm{th}}$, then $r_{1}>k\cdot r_{\mathrm{th}}.$*


**Proof.** Since $r_{\mathrm{th}}=\frac{\ln p_{\mathrm{th}}}{-k\lambda }$, it is true that $e^{-\lambda kr_{\mathrm{th}}}=p_{\mathrm{th}}$. Now assume that $r_{1}$ satisfies the assumption of this lemma. Then,
$$\begin{array}{c} 1-\left(1-e^{-\lambda kr_{\mathrm{th}}}\right)\left(1-e^{-\lambda \sqrt{3}kr_{\mathrm{th}}}\right)\left(1-e^{-\lambda \sqrt{3}kr_{\mathrm{th}}}\right)>1-\left(1-e^{-\lambda kr_{\mathrm{th}}}\right)=e^{-\lambda kr_{\mathrm{th}}}=p_{\mathrm{th}}\\ =1-\left(1-e^{-\lambda r_{1}}\right)\left(1-e^{-\lambda \sqrt{3}r_{1}}\right)\left(1-e^{-\lambda \sqrt{3}r_{1}}\right). \end{array}$$Hence $r_{1}>k\cdot r_{\mathrm{th}}$. □

## 5. Our *k*-Layer Coverage Scheme

### 5.1. The k=1 Case

Our 1-layer coverage scheme deploys nodes according to a “pseudo” triangular pattern deployment, in which neighboring nodes will be regularly separated by a distance of $\sqrt{3}r_{1}$ (i.e., $r_{2}$) except that some of them will be separated by a distance of $\leq \;\frac{\sqrt{3}r_{1}}{2}$ or $\leq \;\frac{3r_{1}}{2}$ and these exceptions only occur at the boundary of the ROI. See Figure 6 for an illustration.

Let *L* and *H* be the length and the height of the ROI, respectively. We assume that the ROI’s lower left corner is located at coordinate (0,0). Then, the coordinates (i.e., locations) of nodes will be calculated row-by-row, with row 1 containing (0,0), and row *ℓ* containing the upper boundary of the ROI. Our full 1-layer coverage scheme is shown in Algorithm 2. Since all nodes have to be within the ROI, we may need to adjust the locations of the following nodes:the *first* node in each even-numbered row (handled by line 6 of Algorithm 2),the *last* node in each row (handled by line 9 of Algorithm 2), and *all* nodes in row *ℓ* (handled by lines 11–18 of Algorithm 2).


**Algorithm 2:**
**Input:** The $r_{1}$ derived by Algorithm 1, the length *L* of the ROI, and the height *H* of the ROI.
**Output:** The designated locations of nodes.
1:$r_{2}\leftarrow \sqrt{3}r_{1}$; $\ell \leftarrow \left\lceil \frac{2H}{3r_{1}}\right\rceil +1$; $n_{1}\leftarrow \left\lceil \frac{L}{r_{2}}\right\rceil +1$; $n_{2}\leftarrow \left\lceil \frac{2L-r_{2}}{2r_{2}}\right\rceil +2$;2:**for**i=1 to ℓ−1
**do**   //row 1, row 2, *…*, row (ℓ−1)3:    **if** (*i* is odd) **then**4:        output the location ($j\cdot r_{2},\frac{3(i-1)}{2}r_{1})$ for each j=0 to $n_{1}-1$;5:    **else**   //*i* is even6:        output the location ($0,\frac{3(i-1)}{2}r_{1})$;   //location of the first node in this row7:        output the location ($\frac{2j+1}{2}r_{2},\frac{3(i-1)}{2}r_{1})$ for each j=0 to $n_{2}-2$;8:    **end if**9:    output the location ($L,\frac{3(i-1)}{2}r_{1})$;   //location of the last node in this row10:**end for**11://row *ℓ* (i.e., the last row)12:**if** (*ℓ* is odd) **then**13:    output the location ($j\cdot r_{2},H)$ for each j=0 to $n_{1}-1$;14:**else**   //*ℓ* is even15:    output the location (0,H);   //location of the first node in row *ℓ*16:    output the location ($\frac{2j+1}{2}r_{2},H)$ for each j=0 to $n_{2}-2$;17:**end if**18:output the location (L,H);   //location of the last node in row *ℓ*


Let *ℓ*, $n_{1}$, and $n_{2}$ denote the number of rows, the number of nodes deployed in an odd-numbered row, and the number of nodes deployed in an even-numbered row by Algorithm 2, respectively. Three claims are made.

**Claim 1.**$\ell =\left\lceil \frac{2H}{3r_{1}}\right\rceil +1$.

**Proof.** This claim follows from the fact that the distance between row *i* and row (i+1), 1≤i<ℓ−1, is exactly $\frac{3}{2}r_{1}$, and the distance between row (ℓ−1) and row *ℓ* is $\leq \;\frac{3}{2}r_{1}$. □

**Claim 2.**$n_{1}=\left\lceil \frac{L}{r_{2}}\right\rceil +1$.

**Proof.** Consider an arbitrary odd-numbered row and let node 1, node 2, *…*, node $n_{1}$ be nodes in this row. Then, the distance between node *i* and node (i+1), $1\leq i<n_{1}-1$, is exactly $r_{2}$, and the distance between node $\left(n_{1}-1\right)$ and node $n_{1}$ is $\leq \;r_{2}$. Hence the claim. □

**Claim 3.**$n_{2}=\left\lceil \frac{2L-r_{2}}{2r_{2}}\right\rceil +2$.

**Proof.** Consider an arbitrary even-numbered row and let node 1, node 2, *…*, node $n_{2}$ be nodes in this row. Then, the distance between node 1 and node 2 is $\frac{r_{2}}{2}$, the distance between node *i* and node (i+1), $2\leq i<n_{2}-1$, is exactly $r_{2}$, and that between node $\left(n_{2}-1\right)$ and node $n_{2}$ is $\leq \;\frac{r_{2}}{2}$. Hence the claim. □

We now are ready to prove Theorem 1.

**Proof.** It can be easily observed from Figure 6 that every location inside the ROI is within zone 1 of at least one node. Thus, to prove this theorem, it suffices to prove that every location inside the ROI is within zones 1–2 of at least three nodes. This assertion can be easily verified by using Figure 7 and hence we have this theorem. □

### 5.2. The General Case

To achieve *k*-layer coverage for k>1, we begin with the 1-layer coverage and duplicate *k* sensor nodes on each designated location of the 1-layer coverage. See Figure 8 for an illustration. See also Appendix A and Figure A1 for possible improvements. The following corollary follows from Claims 1∼3.

**Corollary** **2.**
*The number N of nodes deployed by our k-layer coverage scheme equals to*
$$N=\left\{\begin{array}{cc} k\cdot \left(n_{1}+n_{2}\right)\left\lfloor \frac{\ell }{2}\right\rfloor +k\cdot n_{1} & if\;\ell \;is\;odd,\\ k\cdot \left(n_{1}+n_{2}\right)\left\lfloor \frac{\ell }{2}\right\rfloor & if\;\ell \;is\;even. \end{array}\right.$$


## 6. Experimental Results

Our experimental results are obtained by using Visual C++ programming language under the environment of a 64-bit personal computer with win 10 operating system. Parameters used in our experimental results are listed in Table 3.

Our experimental results are shown in Table 4, Table 5 and Table 6. These results demonstrate $r_{\mathrm{th}}$ used in *k*-threshold coverage [7] and $r_{1}$ used in our *k*-layer coverage. Recall that both $r_{\mathrm{th}}$ and $r_{1}$ are served as a substitution radius for the original sensing radius $r_{s}$. As can be observed in Table 4, Table 5 and Table 6, $r_{\mathrm{th}}$ is much smaller than $r_{s}$ and $r_{1}$ is more reasonable. Table 4, Table 5 and Table 6 also demonstrate the number of nodes required by *k*-threshold coverage [7] and our *k*-layer coverage. It is not difficult to see that the number of nodes required by *k*-threshold coverage [7] is quite large, especially when *k* is large. On the other hand, the number of nodes required by our *k*-layer coverage is much less.

## 7. Concluding Remarks

This paper considers the multilevel (*k*) coverage problem in WSNs with uncertain properties (i.e., under the probabilistic sensing model). We find that the nature of the previous *k*-threshold coverage [7] makes its substitution radius $r_{\mathrm{th}}$ tend to be much smaller than the original sensing range $r_{s}$ and therefore will use too many nodes. We thus try to propose a more reasonable solution and yield the *k*-layer coverage. In *k*-layer coverage, each node’s sensing region is regarded as the composition of zone 1, zone 2, and zone 3. Although regarding each node’s sensing region as the composition of several zones has been proposed in the literature, to the best of our knowledge, we are the first ones to use the “zone 1 and zone 1–2” strategy to solve the probabilistic multilevel (*k*) coverage problem. By using such a “zone 1 and zone 1–2” strategy, our *k*-layer coverage achieves a better coverage quality and uses less nodes. A preliminary version of this paper was in [32]; see also [33]. One future work for this research is to extend *k*-layer coverage to the probabilistic sensing model in (Equation 4) and to more realistic regions of interest. Another future work is to explore the relationship between coverage and energy consumption.

## Figures and Tables

**Figure 1 sensors-20-01831-f001:**
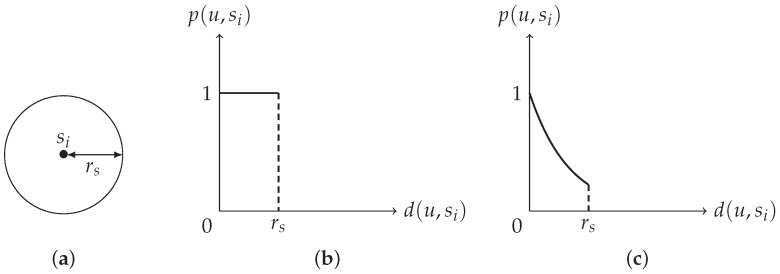
(**a**) Disk shape. (**b**) Binary sensing model. (**c**) Probabilistic sensing model.

**Figure 2 sensors-20-01831-f002:**
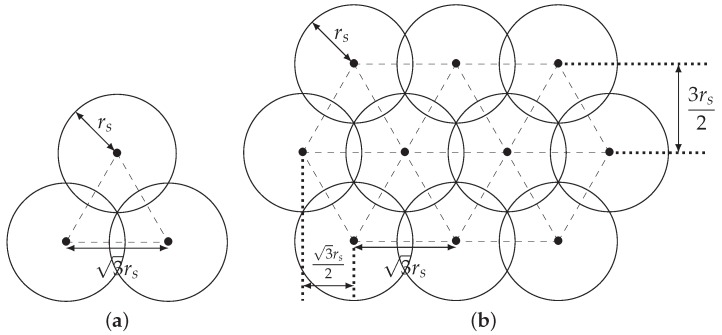
Triangular pattern: (**a**) three neighboring nodes, (**b**) the general case.

**Figure 3 sensors-20-01831-f003:**
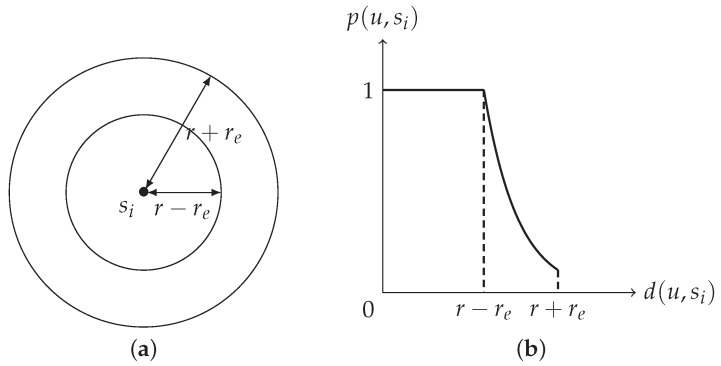
(**a**) $r-r_{e}$ and $r+r_{e}$. (**b**) The generalized probabilistic sensing model.

**Figure 4 sensors-20-01831-f004:**
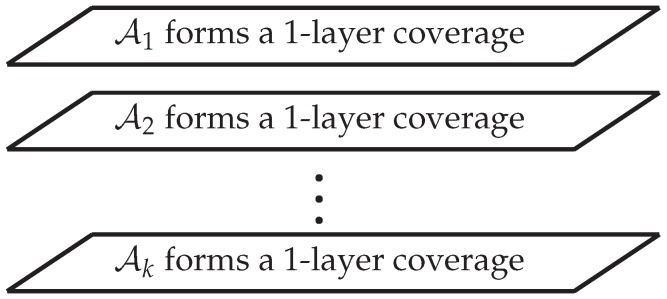
The concept of *k*-layer coverage.

**Figure 5 sensors-20-01831-f005:**
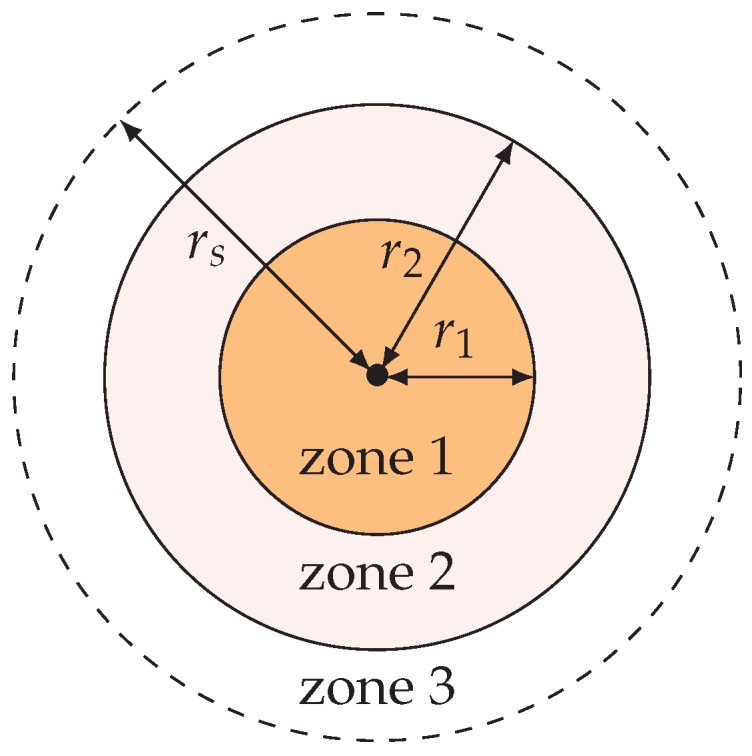
Zone 1, zone 2, zone 3, $r_{1}$, $r_{2}$, and $r_{s}$. We take $r_{2}=\sqrt{3}r_{1}$.

**Figure 6 sensors-20-01831-f006:**
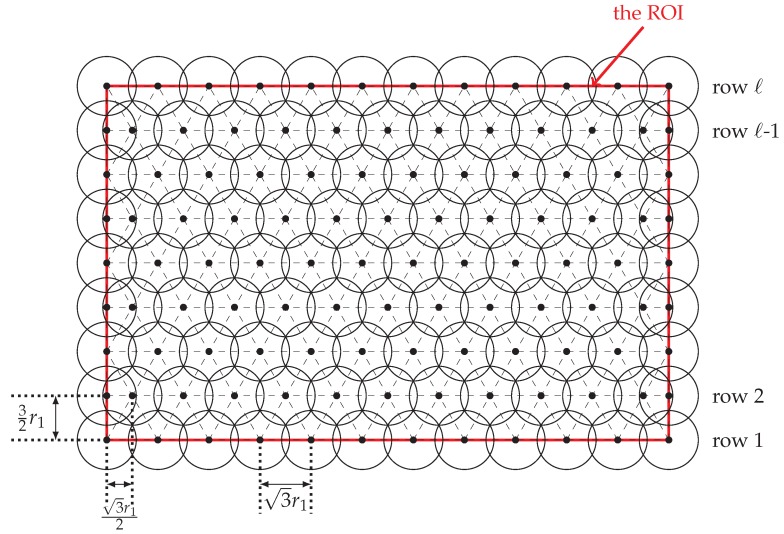
Sensor placement in 1-layer coverage scheme; notice that only zone 1 of each node is shown in this figure. This ROI has $\sqrt{3}r_{1}\;|\;L$ and $\frac{3r_{1}}{2}\;|\;H$ and therefore no adjustment for rows 1,3,…,ℓ.

**Figure 7 sensors-20-01831-f007:**
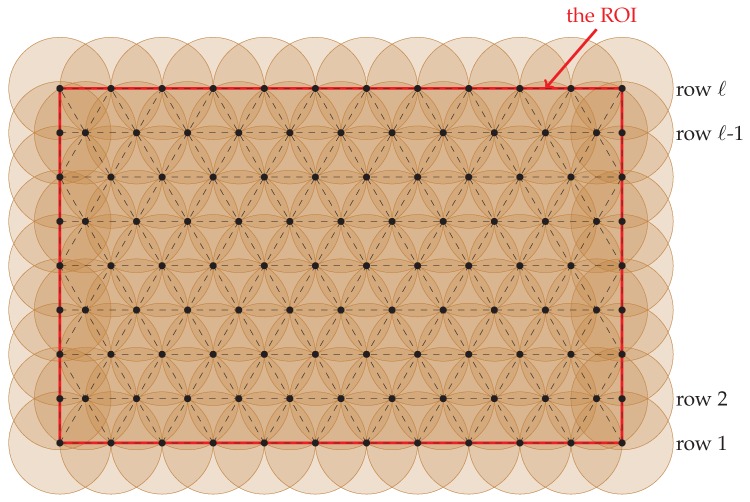
Sensor placement in 1-layer coverage scheme; notice that only zone 1–2 of each node is shown in this figure.

**Figure 8 sensors-20-01831-f008:**
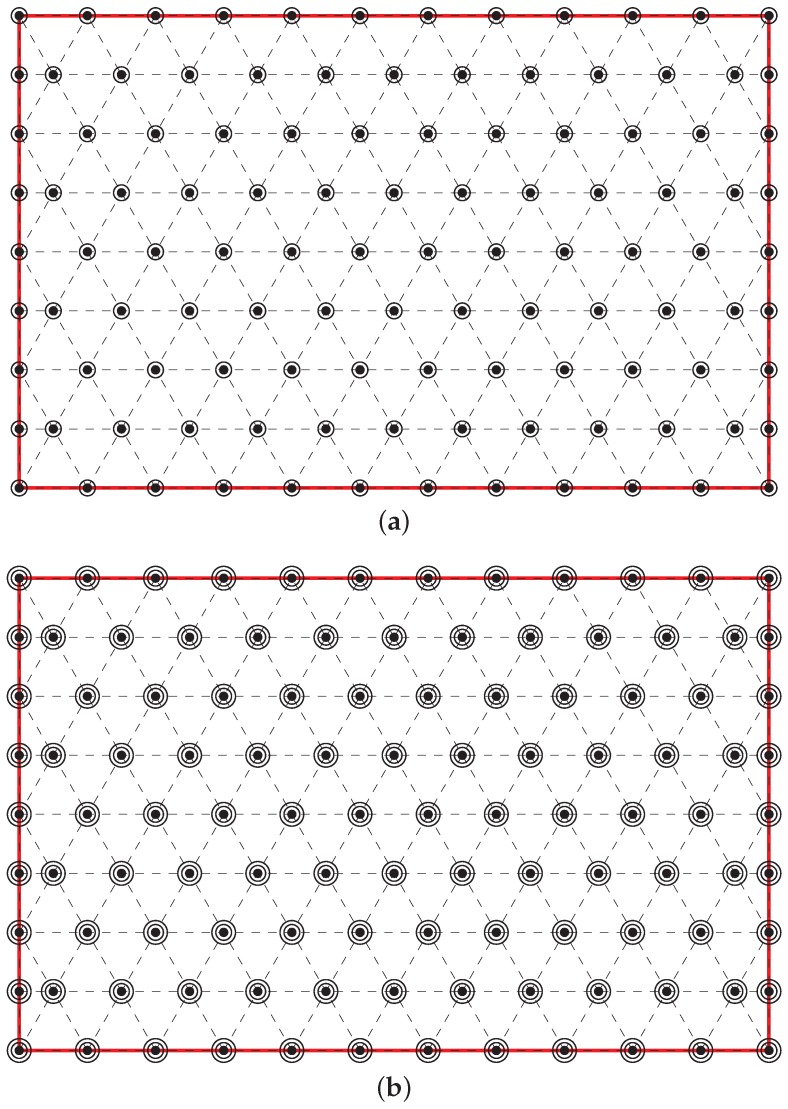
(**a**) 2-layer coverage. (**b**) 3-layer coverage.

**Table 1 sensors-20-01831-t001:** Scheme used by [7]: “*k*” means “to make the ROI *k*-covered”, “duplicate” means “duplicate scheme”, “interpolation” means “interpolation scheme”, and “tri. pattern” means “triangular pattern”.

*k*	Case $r_{c}\leq \frac{\sqrt{3}}{2}r_{s}$	Case $\frac{\sqrt{3}}{2}r_{s}<r_{c}\leq \frac{2+\sqrt{3}}{3}r_{s}$	Case $\frac{2+\sqrt{3}}{3}r_{s}<r_{c}<\sqrt{3}r_{s}$	Case $r_{c}\geq \sqrt{3}r_{s}$
k=1	duplicate	duplicate	duplicate	duplicate and tri. pattern
k=2	duplicate	duplicate	duplicate	duplicate and tri. pattern
k≥3	interpolation	interpolation	duplicate	duplicate and tri. pattern

**Table 2 sensors-20-01831-t002:** Definitions of *k*-coverage: $A=\{s_{1},s_{2},\ldots ,s_{n}\}$ is the set of sensor nodes deployed in the ROI and “det. prob.” means “detection probability”.

*k*-Coverage by	Definition	det. prob. *p* Given by
*k*-expectation [18]	$\sum_{s_{i}\in A}p\left(u,s_{i}\right)\geq k$	Equation (Equation 4)
*k*-threshold [7]	$\prod_{s_{i}\in A^{\prime }}p\left(u,s_{i}\right)\geq p_{\mathrm{th}},\;\mathrm{for}\;\mathrm{some}\;A^{\prime }\subseteq A\;\mathrm{with}\;\left|A^{\prime }\right|=k$	Equation (Equation 1)
*k*-layer (this paper)	see Definition 1	Equation (Equation 1)

**Table 3 sensors-20-01831-t003:** Parameters used: m denote meters.

Size of the ROI	1000 m × 1000 m
sensing range $r_{s}$	30 m
error tolerance ϵ used in Algorithm 1	$10^{-6}$
sensor-dependent parameter λ used in (Equation 1)	either 0.05 or 0.08

**Table 4 sensors-20-01831-t004:** The number of nodes for k=1 coverage: “# nodes” means “the number of nodes required”.

	λ=0.05				λ=0.08			
	**k-Threshold [7]**		**Our k-Layer**		**k-Threshold [7]**		**Our k-Layer**	
$p_{\mathrm{th}}$	$r_{\mathrm{th}}\;$	**# nodes**	$r_{1}\;\;\;$	**# nodes**	$r_{\mathrm{th}}\;$	**# nodes**	$\;\;\;r_{1}\;\;\;$	**# nodes**
0.7	7.133	7790	15.685	1672	4.458	19,781	9.803	4200
0.8	4.462	19,781	12.391	2640	2.789	50,369	7.744	6688
0.9	2.107	87,768	8.749	5226	1.317	223,520	5.468	13,161

**Table 5 sensors-20-01831-t005:** The number of nodes for k=3 coverage: “# nodes” means “the number of nodes required”.

	λ=0.05				λ=0.08			
	**k-Threshold [7]**		**Our k-Layer**		**k-Threshold [7]**		**Our k-Layer**	
$p_{\mathrm{th}}$	$r_{\mathrm{th}}\;$	**# nodes**	$r_{1}\;\;\;$	**# nodes**	$r_{\mathrm{th}}\;$	**# nodes**	$\;\;\;r_{1}\;\;\;$	**# nodes**
0.7	2.377	206,424	15.685	5016	1.486	526,500	9.803	12,600
0.8	1.487	526,500	12.391	7920	0.929	1,343,811	7.744	20,064
0.9	0.702	2,350,872	8.749	15,678	0.439	6,005,520	5.468	39,483

**Table 6 sensors-20-01831-t006:** The number of nodes for k=5 coverage: “# nodes” means “the number of nodes required”.

	λ=0.05				λ=0.08			
	**k-Threshold [7]**		**Our k-Layer**		**k-Threshold [7]**		**Our k-Layer**	
$p_{\mathrm{th}}$	$r_{\mathrm{th}}\;$	**# nodes**	$r_{1}\;\;\;$	**# nodes**	$r_{\mathrm{th}}\;$	**# nodes**	$\;\;\;r_{1}\;\;\;$	**# nodes**
0.7	1.426	952,070	15.685	8360	0.891	2,433,750	9.803	21,000
0.8	0.892	2,430,505	12.391	13,200	0.557	6,217,600	7.744	33,440
0.9	0.421	10,881,000	8.749	26,130	0.263	27,857,950	5.468	65,805

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
