# Peer review of "An Effective Sensor Deployment Scheme that Ensures Multilevel Coverage of Wireless Sensor Networks with Uncertain Properties"

_sensors, 2020, doi:10.3390/s20071831_

Round 1
Reviewer 1 Report
Authors propose a sensor deployment scheme that minimizes the number of deployed sensor nodes while ensuring good coverage qualities using a novel “zone 1 & zone 1–2” strategy
Suggestions:
- What software/hardware is used to obtain the experimental results?
- Experimental results improve k-threshold in reference [20] (2008), but it is necessary to compare with more recent papers, of similar characteristics (deterministic, homogeneous, static ...)
- Algorithms should be verified in a real controllable environment.
Author Response
Please see the attache file. Thank you.

Reviewer 2 Report
The paper is well written and easy to follow. It is mathematically sound, with a clear analytical framework.
The paper proposes a definition of k-coverage under the probability sensing model. Then, they propose a new mechanism, called zone 1 and zone 1-2, to provide k-coverage while reducing the number of nodes in the region of interest.
The authors develop an algorithm to find the adequate value of r1 which can be found after only 18 iterations for an error of 10^-6.
My comments are as follows:
1) The authors mention that in the literature, there is not a single definition for the decay sensing function in the probabilistic sensing model, equation (1) of the manuscript. However, they seem to use a particular definition for their proposal. Since there is not a single definition for this function, I suggest that the authors provide results using a different function. Since the mathematical analysis may be hard to provide, simulation results or approximated results should be enough or even a clear discussion on the impact of this function.
2) The authors mention a relation between coverage and energy consumption. However, they do not provide further details regarding this relationship. The authors should provide a more detailed discussion on this topic and if possible, numerical results showing how the coverage impacts the energy consumption of the WSN.
Reviewer 3 Report
Please find below my comments.
- Separate the related work from introduction Section and add a Related work section.
- Add comparative table of Related work by clearly mentioning problem, aim, proposed solution, pros and cons.
- These papers also presented solutions to increase the coverage and reliability. Discuss and cite it. a. “An analytical approach to opportunistic transmission under rayleigh fading channels ,” Sage Hindawi International Journal of Distributed Sensor Networks, vol. 2015, pp. 1-8, November 2015. b. “Heuristic approach to select opportunistic routing forwarders (HASORF) to enhance throughput for wireless sensor networks,” Hindawi Journal of Sensors, vol. 2015, pp. 1-10, June 2015.
- Add detail simulation parameters in tabular form.
Author Response
Please see the attache file. Thank you.

Round 2
Reviewer 1 Report
- Authors should compare the results of their proposal with others in the same field.
- A few possible examples:
- Rezaee, Abbas Ali et al. “Coverage Optimization in Wireless Sensor Networks Using Gravitational Search Algorithm.” (2019).
- R. Özdağ, "The solution of the k-coverage problem in Wireless Sensor Networks," 2016 24th Signal Processing and Communication Application Conference (SIU), Zonguldak, 2016, pp. 873-876.
- Q. Liu, "K-Coverage Reliability Evaluation for Wireless Sensor Networks Using 2 Dimensional k / r×s / m×n:F System," 2018 12th International Conference on Reliability, Maintainability, and Safety (ICRMS), Shanghai, China, 2018, pp. 83-87.
- M. Elhoseny, A. Tharwat, A. Farouk and A. E. Hassanien, "K-Coverage Model Based on Genetic Algorithm to Extend WSN Lifetime," in IEEE Sensors Letters, vol. 1, no. 4, pp. 1-4, Aug. 2017, Art no. 7500404.
- H. M. Ammari and S. K. Das, "Centralized and Clustered k-Coverage Protocols for Wireless Sensor Networks," in IEEE Transactions on Computers, vol. 61, no. 1, pp. 118-133, Jan. 2012.
- Recommended reading:
M. Farsi, M. A. Elhosseini, M. Badawy, H. Arafat Ali and H. Zain Eldin, "Deployment Techniques in Wireless Sensor Networks, Coverage and Connectivity: A Survey," in IEEE Access, vol. 7, pp. 28940-28954, 2019.
Author Response
Dear reviewer:
Please see the attached file. Thank you.

Reviewer 3 Report
Authors did not address all of my comments.
Author Response

(The authors gave the same response as above.)

Round 3
Reviewer 1 Report
The questions raised have been adequately answered.
Author Response
Thank you very much.
Reviewer 3 Report
I am satisfied with the revised version.
Author Response
Thank you very much.